# Learning from Label Proportions by Learning with Label Noise

**Jianxin Zhang, Yutong Wang, and Clayton Scott**
Electrical Engineering and Computer Science
University of Michigan
Ann Arbor, MI 48109
`{jianxinz, yutongw, clayscot}@umich.edu`

## Abstract

Learning from label proportions (LLP) is a weakly supervised classification problem where data points are grouped into bags, and the label proportions within each bag are observed instead of the instance-level labels. The task is to learn a classifier to predict the labels of future individual instances. Prior work on LLP for multi-class data has yet to develop a theoretically grounded algorithm. In this work, we propose an approach to LLP based on a reduction to learning with label noise, using the forward correction (FC) loss of Patrini et al. [30]. We establish an excess risk bound and generalization error analysis for our approach, while also extending the theory of the FC loss which may be of independent interest. Our approach demonstrates improved empirical performance in deep learning scenarios across multiple datasets and architectures, compared to the leading methods.

## 1 Introduction

In the weakly supervised problem of *learning from label proportions* (LLP), the learner is presented with bags of instances, where each bag is annotated with the proportions of the different classes in the bag. The learner's objective is to produce a classifier that accurately assigns labels to individual instances in the future. LLP arises in various applications including high energy physics [7], election prediction [45], computer vision [4, 20], medical image analysis [2], remote sensing [8], activity recognition [32], and reproductive medicine [12].

To date, most methods for LLP have addressed the setting of binary classification [50, 36, 39, 34, 41, 44, 24, 37, 38], although multiclass methods have also recently been investigated [9, 22, 46]. The dominant approach to LLP in the literature is "label proportion matching": train a classifier to accurately reproduce the observed label proportions on the training data, perhaps with additional regularization. In the multiclass setting, the Kullback-Leibler (KL) divergence between the observed and predicted label proportions is adopted by the leading approaches to assess proportion matching. Unfortunately, while matching the observed label proportions is intuitive and can work well in some settings, it has little theoretical basis [50, 38], especially in the multiclass setting, and there are natural settings where it fails [50, 39].

Recently, Scott and Zhang [39] demonstrated a principled approach to LLP with performance guarantees based on a reduction to learning with label noise (LLN) in the binary setting. Their basic strategy was to pair bags, and view each pair of bags as an LLN problem, where the observed label proportions are related to the "label flipping" or "noise transition" probabilities. Using an existing technique for LLN based on loss correction, which allows the learner to train directly on the noisy data, they formulated an overall objective based on a (weighted) sum of objectives for each pair of bags. They established generalization error analysis and consistency for the method, and also showed that in the context of kernel methods, their approach outperformed the leading kernel methods.

36th Conference on Neural Information Processing Systems (NeurIPS 2022).

The objective of the present paper is to develop a theoretically grounded and practical approach to multiclass LLP, drawing inspiration from Scott and Zhang [39]. The primary challenge stems from the fact that Scott and Zhang [39] employed the so-called "backward correction" loss, which solves LLN by scaling the *output* of a loss function of interest according to the noise transition probabilities [28, 30, 35]. While this loss correction was demonstrated to work well for kernel methods in a binary setting, Patrini et al. [30] introduced an alternative loss correction that performs better empirically in deep learning settings (see also [53]). They proposed the "forward correction" loss, which scales the *inputs* to a loss function of interest according to the noise transition probabilities. Patrini et al. [30] find that backward correction "does not seem to work well in the low noise regime," and is "a linear combination of losses" with "coefficients that can be far [apart] by orders of magnitude " which "makes the learning harder".

The present work is thus inspired by Scott and Zhang [39] but uses the forward correction (FC) loss in a multiclass setting. This requires a number of technical modifications to the arguments of Scott and Zhang [39]. Most notably, it now becomes necessary to demonstrate that the FC loss is *calibrated* with respect to the 0-1 loss, a critical property needed for showing consistency. Such analysis is inherently not needed when using the backward correction, where the target excess risk is *proportional to* the surrogate excess risk (from which calibration follows trivially). Furthermore, Scott and Zhang [39] does not require analysis of proper composite losses, which are needed in the FC framework. Finally, the multiclass setting involves new estimation challenges not present in the binary case. These factors mean that our work is not a straightforward extension of Scott and Zhang [39]. Indeed, the authors of a recent report acknowledge that it is "difficult to extend [the method of Scott and Zhang [39]] to multiclass classification" [16].

**Additional related work:** Much work on LLP has focused on learning specific types of models, including support vector machines [36, 50, 47, 33, 5, 19, 40], probabilistic models [18, 13, 45, 32, 12], random forests [41], neural networks [21, 1, 9, 22, 46], and clustering-based models, [3, 44]. Many of these works develop learning criteria that are specific to the model being learned.

On the theoretical front, Quadrianto et al. [34] and Patrini et al. [30] initiated the learning theoretic study of LLP, introducing Rademacher style bounds for linear methods, but they do not address consistency w.r.t. a classification performance measure. Yu et al. [51] provides support for label proportion matching but only under the assumption that the bags are very pure. Saket [37] studies learnability of linear threshold functions. Recently Saket et al. [38] introduced a condition under which label proportion matching does provably well w.r.t. a squared error loss in the binary setting, and developed an associated algorithm. This method does not scale easily to large datasets, and further requires knowledge of how bags are grouped according to different bag-generating distributions.

A handful of recent papers have studied multiclass LLP in deep learning scenarios. Dulac-Arnold et al. [9] study the KL loss for label proportion matching, and a variant based on optimal transport. Liu et al. [22, 23] examine an approach based on generative adversarial models. Tsai and Lin [46] study the use of a regularizer derived from semi-supervised learning. One challenge common to these approaches is that their implementations employ mini-batches of bags, which becomes computationally prohibitive for large bag sizes when the batch size is still very small, e.g., 2 or 3 bags. In contrast, our approach avoids this issue. Finally, a recent technical report presents a risk analysis for multiclass LLP under the assumption of fixed bag size, which we do not require [16]. Their method is not tractable for large bag sizes in which case they approximate their objective "using the bag-level loss proposed in the existing research."

**Contributions and Outline:** Our contributions and the paper structure are summarized as follows. In Section 2, we review the FC loss as a solution to LLN. In Section 3, we extend the theory of the FC loss for LLN. In particular, we show that the FC loss is "uniformly calibrated" with respect to the 0-1 loss using the framework of Steinwart [43], establish an excess risk bound, and determine an explicit lower bound on the calibration function in terms of the noise transition matrix. In Section 4, we extend the results of Section 3 to the setting with multiple noise transition matrices, which form the basis of our approach to LLP. In particular, we establish an excess risk bound and generalization error analysis for learning with multiple noise transition matrices, which in turn enables proofs of consistency. In Section 5, we state the probabilistic model for reducing LLP to LLN with multiple different noise transition matrices and present the LLPFC algorithms. Experiments with deep neural networks are presented in Section 6, where we observe that our approach outperforms competing methods by a substantial margin. Proofs appear in the supplemental material.

## 2 Learning with Label Noise and the Forward Correction Loss

This section sets notation and introduces the FC loss as a solution to learning with label noise. Let $\mathcal{X}$ be the feature space and $\mathcal{Y} = \{1, 2, \ldots, C\}$ be the label space, $C \in \mathbb{N}$. We define the $C$-simplex as $\Delta^C = \{p \in \mathbb{R}^C : p_i \geq 0, \forall i = 1, 2, \ldots, C, \sum_{i=1}^C p_i = 1\}$ and denote its interior by $\mathring{\Delta}^C$. Let $P$ be a probability measure on the space $\mathcal{X} \times \mathcal{Y}$.

Viewing $P$ as the "clean" probability measure, a noisy probability measure with label-dependent label noise can be constructed from $P$ in terms of a $C \times C$ column-stochastic matrix $T$, referred to as the *noise transition matrix*. Formally, we define a measure $\bar{P}_T$ on $\mathcal{X} \times \mathcal{Y} \times \mathcal{Y}$ by requiring $\forall$ events $\mathcal{A} \subset \mathcal{X}, \bar{P}_T(\mathcal{A} \times \{i\} \times \{j\}) = P(\mathcal{A} \times \{i\}) t_{j,i}$ where $t_{j,i}$ is the element at the $j$-th row and $i$-th column of $T$. Let $(X, Y, \tilde{Y})$ have joint distribution $\bar{P}_T$ where $X$ is the feature vector, $Y$ is the "clean" label, and $\tilde{Y}$ is the "noisy" label. Thus the element of $T$ at row $i$ and column $j$ is $t_{i,j} = \bar{P}_T(\tilde{Y} = i | Y = j)$. In addition, $P$ is the marginal distribution of $(X, Y)$. Define $P_T$ to be the marginal distribution of $(X, \tilde{Y})$. Let $\mathcal{F}$ be the collection of all measurable functions from $\mathcal{X}$ to $\Delta^C$.

The existence of a regular conditional distribution is guaranteed by the Disintegration Theorem (*e.g.* Theorem 6.4 in Kallenberg [14]) under suitable properties (*e.g.* when $\mathcal{X}$ is a Radon space). While the existence of regular conditional probability is beyond the scope of this paper, we assume fixed regular conditional distributions for $Y$ and $\tilde{Y}$ given $X$ exist, denoted by $P(\cdot \mid \cdot) : \mathcal{Y} \times \mathcal{X} \to [0, 1]$ and $P_T(\cdot \mid \cdot) : \mathcal{Y} \times \mathcal{X} \to [0, 1]$, respectively. Given $x \in \mathcal{X}$, we define the probability vectors $\eta(x) = [P(1 \mid x), \ldots, P(C \mid x)]^{tr}$ and $\eta_T(x) = [P_T(1 \mid x), \ldots, P_T(C \mid x)]^{tr}$ where we use $tr$ to denote transposition. It directly follows that $\eta_T(x) = T\eta(x)$.

We use $\mathbb{R}_+$ to denote the positive real numbers. The goal of LLN is to learn a classifier that optimizes a performance measure defined *w.r.t.* $P$, given access to corrupted training data $(X_i, \tilde{Y}_i) \overset{i.i.d.}{\sim} P_T$. In this work we assume $T$ is known or can be estimated, as is the case when we apply LLN techniques to LLP (see Section 5). A more formal formulation of LLP is given in Section 5.

When attempting to minimize the risk associated to the 0-1 loss and the clean distribution $P$, it is common to employ a smooth or convex surrogate loss. For LLN problems, the idea of a *loss correction* is to modify the surrogate loss so that when optimized using the *noisy* data, it still achieves the desired goal. Below, we introduce the forward correction loss, before which we need to define inner risk and proper loss. For this purpose we focus on loss functions of the form $L : \Delta^C \times \mathcal{Y} \to \mathbb{R}$.

**Definition 1.** *Let* $L : \Delta^C \times \mathcal{Y} \to \mathbb{R}$ *be a loss function. The **inner $L$-risk** at $x$ with probability measure $P$ is* $\mathcal{C}_{L,P,x} : \Delta^C \to \mathbb{R}, \ \mathcal{C}_{L,P,x}(q) := \mathbb{E}_{Y \sim P(\cdot|x)} L(q, Y)$. *The **minimal inner $L$-risk** at $x$ with a probability measure $P$ is* $\mathcal{C}^*_{L,P,x} := \inf_{q \in \Delta^C} \mathcal{C}_{L,P,x}(q)$.

**Definition 2.** $\ell : \Delta^C \times \mathcal{Y} \to \mathbb{R}$ *is a **proper loss** if $\forall$ probability measures $P$ on $\mathcal{X} \times \mathcal{Y}$, $\forall x \in \mathcal{X}, \mathcal{C}^*_{\ell,P,x} = \mathcal{C}_{\ell,P,x}(\eta(x))$, and a proper loss is called **strictly proper** if the minimizer of $\mathcal{C}_{\ell,P,x}$ is unique for all $x \in \mathcal{X}$.*

Commonly used proper losses include the *log loss* $\ell^{log}(q, c) = -\log q_c$, the *square loss* $\ell^{sq}(q, c) = \sum_{c'=1}^C (\mathbb{1}_{c=c'} - q_{c'})^2$, and the *0-1 loss* $\ell^{01}(q, c) = \mathbb{1}_{c \neq \min\{\arg\max_j q_j\}}$, among which only the log loss and the square loss are strictly proper [49]. Here $\mathbb{1}$ denotes the indicator function. Note that it is common to compose proper losses with inverted link functions, leading to familiar losses like the cross-entropy loss. Such losses are discussed further in Section 4.

We are now ready to introduce the forward correction loss.

**Definition 3.** *Let $\ell$ be a strictly proper loss and let $T$ be a noise transition matrix. Define the **forward correction loss** of $\ell$ as $\ell_T : \Delta^C \times \mathcal{Y} \to \mathbb{R}, \ \ell_T(q, c) := \ell(Tq, c)$.*

It follows from the definition that, if $T$ is invertible, then the inner $\ell_T$-risk under the distribution $P_T$ has a unique minimizer $\eta(x)$. Next we introduce $L$-risk and $L$-Bayes risk associated with a loss $L$.

**Definition 4.** *Let $L : \Delta^C \times \mathcal{Y} \to \mathbb{R}$ and $P$ be a probability measure. Define the **$L$-risk** of $f$ with distribution $P$ to be $\mathcal{R}_{L,P} : \mathcal{F} \to \mathbb{R}, \ \mathcal{R}_{L,P}(f) := \mathbb{E}_P [L(f(X), Y)]$ and the **$L$-Bayes risk** to be $\mathcal{R}^*_{L,P} := \inf_{f \in \mathcal{F}} \mathcal{R}_{L,P}(f)$.*

We call $\mathcal{R}_{L,P}(f) - \mathcal{R}^*_{L,P}$ the *excess $L$-risk* of $f$ under distribution $P$. Given a proper loss $\ell$, Theorem 2 of Patrini et al. [30] establishes Fisher consistency of the FC loss, meaning the minimizer of $\ell$-risk

under the clean distribution $P$ is the same as the minimizer of $\ell_T$-risk under noisy distribution $P_T$: $\arg\min_{f \in \mathcal{F}} \mathcal{R}_{L,P}(f) = \arg\min_{f \in \mathcal{F}} \mathcal{R}_{\ell_T,P_T}(f)$. Next, we present a stronger result relating the excess $\ell_T$-risk under the noisy distribution $P_T$ to the excess 0-1 risk under the clean distribution $P$.

## 3   Calibration Analysis for the Forward Correction Loss

Our objective in this section is to show that when $L$ is the 0-1 loss and $\ell$ is a continuous strictly proper surrogate loss, there exists a strictly increasing, invertible function $\theta$ with $\theta(0) = 0$ such that $\forall f \in \mathcal{F}$ and $\forall$ distributions $P$, $\theta\big(\mathcal{R}_{L,P}(f) - \mathcal{R}^*_{L,P}\big) \le \mathcal{R}_{\ell_T,P_T}(f) - \mathcal{R}^*_{\ell_T,P_T}$. Given such a bound, it follows that consistency $w.r.t$ the surrogate risk implies consistency $w.r.t.$ the target risk. The results in this section are standalone results for the FC loss that may be of independent interest, and will be extended in the next section in relation to LLP. The following theorem guarantees the existence of such function $\theta$, given that $T$ is invertible.

**Theorem 5.** *Let $\ell$ be a continuous strictly proper loss and $T$ be an invertible column-stochastic matrix. Let $L$ be the $0$-$1$ loss. Assume $\mathcal{R}^*_{\ell_T,P_T} < \infty$. Then $\exists \theta : [0,1] \to [0,\infty]$ that is strictly increasing and continuous, satisfying $\theta(0) = 0$, such that $\forall f \in \mathcal{F}, \mathcal{R}_{L,P}(f) - \mathcal{R}^*_{L,P} \le \theta^{-1}\big(\mathcal{R}_{\ell_T,P_T}(f) - \mathcal{R}^*_{\ell_T,P_T}\big)$.*

The function $\theta$ in Theorem 5 depends on $\ell$ and $T$. The following proposition provides a convex lower bound on $\theta$ for the commonly used log loss $\ell^{log}(q,c) = -\log q_c$. Let $M \in \mathbb{R}^{C \times C}$ be a matrix and let $\|\cdot\|$ be a norm on $\mathbb{R}^C$. The *subordinate matrix norm* induced by $\|\cdot\|$ is $\|M\| := \sup_{x \in \mathbb{R}^C : x \neq 0} \frac{\|Mx\|}{\|x\|}$. When $\|\cdot\|$ is the 1-norm on $\mathbb{R}^C$, the induced norm is denoted $\|M\|_1$, referred to as the matrix 1-norm, and can be computed as $\|M\|_1 = \max_{1 \le j \le C} \sum_{i=1}^C |M(i,j)|$ [10].

**Proposition 6.** *Let $T \in \mathbb{R}^{C \times C}$ be an invertible, column-stochastic matrix. Define $\underline{\theta}_T : [0,\infty] \to [0,\infty]$ by $\underline{\theta}_T(\epsilon) = \frac{1}{2} \frac{\epsilon^2}{\|T^{-1}\|_1^2}$. If $L$ is the 0/1 loss, $\ell$ is the log loss, then for all $f \in \mathcal{F}$ and distributions $P$, $\mathcal{R}_{L,P}(f) - \mathcal{R}^*_{L,P} \le \underline{\theta}_T^{-1}\big(\mathcal{R}_{\ell_T,P_T}(f) - \mathcal{R}^*_{\ell_T,P_T}\big) = \sqrt{2}\|T^{-1}\|_1 \sqrt{\mathcal{R}_{\ell_T,P_T}(f) - \mathcal{R}^*_{\ell_T,P_T}}$*

The factor $\|T^{-1}\|_1$ may be viewed as a constant that captures the overall amount of label noise. The more noise, the larger the constant. For example, let $I$ and $N$ be the identity and the all $1/C$'s matrices, respectively. Let $\alpha \in [0,1]$ and $T = (1-\alpha)I + \alpha N$. Thus, $\alpha = 0$ represents the noise-free case and $\alpha = 1$ the noise-only case. It is easy to verify that $T^{-1} = (1-\alpha)^{-1}(I - \alpha N)$ and $\|T^{-1}\|_1 = (1-\alpha)^{-1}(1 + (1 - 2/C)\alpha)$.

## 4   Learning with Multiple Noise Transition Matrices

Our algorithms for LLP, formally stated in subsection 5.4, reduce the problem of LLP to LLN by partitioning bags into groups and modeling each group as an LLN problem. Since each group has its own noise transition matrix, this leads to a new problem that we refer to as learning with multiple noise transition matrices (LMNTM). In this section, we show how to extend the calibration analysis of section 3 to this setting. In addition, we offer a generalization error bound that justifies an empirical risk minimization learning procedure based on a weighted sum of FC losses.

### 4.1   Learning with Multiple Noise Transition Matrices

We first define the LMNTM problem formally. For all $n \in \mathbb{N}$, denote $\mathbb{N}_n = \{1, 2, \dots, n\}$. Consider a clean distribution $P$ on $\mathcal{X} \times \mathcal{Y}$ and noise transition matrices $T_1, T_2, \dots, T_N$. For each $i$ we denote the noisy prior as the $\alpha_i \in \mathring{\Delta}^C$ where, $\forall c \in \mathcal{Y}, \alpha_i(c) = P_{T_i}(\tilde{Y} = c)$. We assume the $\alpha_i$'s are known for theoretical analysis. In practice, $\alpha_i$ is estimable as discussed below. In LMNTM, we observe data points $S = \big\{X_{i,c,j} : i \in \mathbb{N}_N, c \in \mathcal{Y}, j \in \mathbb{N}_{n_{i,c}}\big\}$ where $X_{i,c,j} \overset{iid}{\sim} P_{T_i}(\cdot \mid c)$, and $n_{i,c} \in \mathbb{N}$ is the number of data points drawn from the class conditional distribution $P_{T_i}(\cdot \mid c)$. Assume all $X_{i,c,j}$'s are mutually independent. We make additional remarks on this setting in Section C.1 in the appendix.

## 4.2 A Risk for LMNTM

The following result extends Theorem 5 to LMNTM. It establishes that the risk $\widetilde{R}_{\ell,P,\mathcal{T}}$, which can be estimated from LMNTM training data, is a valid surrogate risk. This type of result is not needed for the backward correction approach of Scott and Zhang [39].

**Theorem 7.** *Let L be the 0-1 loss and $N \in \mathbb{N}$. Consider a sequence of invertible column-stochastic matrices $\mathcal{T} = \{T_i\}_{i=1}^N$ and a continuous strictly proper loss function $\ell$. Let $w = (w_i)_{i=1}^N \in \Delta^N$. Define $\widetilde{R}_{\ell,P,\mathcal{T}} : \mathcal{F} \to \mathbb{R}$ by $\widetilde{R}_{\ell,P,\mathcal{T}}(f) := \sum_{i=1}^N w_i \mathcal{R}_{\ell_{T_i},P_{T_i}}(f)$ and $\widetilde{R}^*_{\ell,P,\mathcal{T}} = \inf_{f \in \mathcal{F}} \widetilde{R}_{\ell,P,\mathcal{T}}(f)$. Assume $\forall i \in \{1,2,\ldots,N\}, \mathcal{R}^*_{\ell_{T_i},P_{T_i}} < \infty$. Then $\exists$ a strictly increasing continuous function $\theta : [0,1] \to [0,\infty]$ with $\theta(0) = 0$ s.t. for all $P$, $\forall f \in \mathcal{F}, \theta\big(\mathcal{R}_{L,P}(f) - \mathcal{R}^*_{L,P}\big) \leq \widetilde{\mathcal{R}}_{\ell,P,\mathcal{T}} - \widetilde{\mathcal{R}}^*_{\ell,P,\mathcal{T}}.$*

The weights $w_i$ allow the user flexibility, for example, to place different weights on noisier or larger subsets of data. Unlike Scott and Zhang [39], however, because the weights appear in both our excess risk bound and generalization error bound, it is not straightforward to optimize them a priori. We discuss weight optimization in detail in Section F in the appendix.

## 4.3 Generalization Error Bound

The aggregate risk $\widetilde{R}_{\ell,P,\mathcal{T}}$ is desirable because it can naturally be estimated from the given data. We propose the empirical risk

$$\hat{\mathcal{R}}_{w,S}(f) = \sum_{i=1}^N w_i \sum_{c=1}^C \frac{\alpha_i(c)}{n_{i,c}} \sum_{j=1}^{n_{i,c}} \ell_{T_i}(f(X_{i,c,j}), c). \tag{1}$$

It should be noted that $\hat{\mathcal{R}}_{w,S}(f)$ is an unbiased estimate of $\widetilde{\mathcal{R}}_{\ell,P,\mathcal{T}}(f)$. Here we establish a generalization error bound for this estimate which builds on Rademacher complexity analysis .

To state the bound, we must first introduce the notion of a *proper composite loss* [49]. This stems from the fact that in practice, a function $f$ outputting values in $\Delta^C$ is typically obtained by composing a $\mathbb{R}^C$-valued function (such as a neural network with $C$ output layer nodes), with another function $\mathbb{R}^C \to \Delta^C$ such as the softmax function. Thus, let $\psi : \mathcal{U} \subset \Delta^C \to \mathcal{V}$ be an invertible function where $\mathcal{V}$ is a subset of a normed space, referred to as an *invertible link function*. Consider $\mathcal{G} \subset \psi \circ \mathcal{F} := \{\psi \circ f : f \in \mathcal{F}\}$, and observe that $\forall g \in \mathcal{G}, \psi^{-1} \circ g \in \mathcal{F}$. In practice, $\psi$ is fixed and we seek to learn $g \in \mathcal{G}$ that leads to an $f \in \mathcal{F}$ with a risk close to the Bayes risk. An example of $\psi^{-1}$ is the softmax function so that $\psi : \mathcal{U} \to \mathcal{V}, \psi_i(p) = \log p_i - \frac{1}{C} \sum_{k=1}^C \log p_k, (\psi^{-1})_i(s) = \frac{e^{s_i}}{\sum_{k=1}^C e^{s_k}}$ where $\mathcal{U}$ is the interior of $\Delta^C$ and $\mathcal{V} = \{s \in \mathbb{R}^C : \sum_{i=1}^C s_i = 0\}$. This motivates the following definition.

**Definition 8.** *Given an invertible link function $\psi : \mathcal{U} \subset \Delta^C \to \mathcal{V}$, we define the **proper composite loss** $\lambda_\ell$ of a proper loss $\ell : \Delta^C \times \mathcal{Y} \to \mathbb{R}$ to be $\lambda_\ell : \mathcal{V} \times \mathcal{Y} \to \mathbb{R}, \ \lambda_\ell(v,c) = \ell\big(\psi^{-1}(v), c\big)$.*

For example, when $\ell$ is the log loss and $\psi^{-1}$ is the softmax function, $\lambda_\ell$ is the cross-entropy (or multinomial logistic) loss. With this notation, we are now able to state our generalization error bound for LMNTM. We study two popular choices of function classes, the reproducing kernel Hilbert space (RKHS) and the multilayer perceptron (MLP). We use $\mathcal{G}_1$ to denote the Cartesian product of $C$ balls of radius R in the RKHS and $\mathcal{G}_2$ to denote a multilayer perceptron with $C$ outputs.

**Definition 9.** *Let $k$ be a symmetric positive definite (SPD) kernel, and let $\mathcal{H}$ be the associated reproducing kernel Hilbert space (RKHS). Assume $k$ is bounded by $K$, meaning $\forall x, \|k(\cdot,x)\|_{\mathcal{H}} \leq K$. Let $\mathcal{G}^k_{K,R}$ denote the ball of radius $R$ in $\mathcal{H}$. Define $\mathcal{G}_1 = \mathcal{G}^k_{K,R} \times \mathcal{G}^k_{K,R} \times \cdots \times \mathcal{G}^k_{K,R}$ ($C$ copies).*

We follow Zhang et al. [54] and define real-valued MLPs inductively:

**Definition 10.** *Define $\mathcal{N}_1 = \{x \to \langle x, v \rangle : v \in \mathbb{R}^d, \|v\|_2 \leq \beta\}$, and for $m > 2$, inductively define $\mathcal{N}_m = \{x \to \sum_{j=1}^d v_j \mu(f_j(x)) : v \in \mathbb{R}^d, \|v\|_1 \leq \beta, f_j \in \mathcal{N}_{m-1}\}$, where $\beta \in \mathbb{R}_+$ and $\mu$ is a 1-Lipschitz activation function. Define an MLP which outputs a vector in $\mathbb{R}^C$ by $\mathcal{G}_2 = \mathcal{N}_m \times \mathcal{N}_m \times \cdots \times \mathcal{N}_m$ ($C$ copies). We additionally assume that the choice of $\mu$ satisfies $\forall m \in \mathbb{N}, 0 \in \mu \circ \mathcal{N}_m$.*

**Theorem 11.** *Let $T_1, T_2, \ldots, T_N$ be invertible column-stochastic matrices. Let $\ell$ be a proper loss such that $\forall i,c$ the function $\lambda_{\ell_{T_i}}(\cdot,c)$ is Lipschitz continuous w.r.t. the 2-norm. Let $S$ be the set of*

*data points as defined in Section 4.1. Assume $\sup_{x \in \mathcal{X}, g \in \mathcal{G}_q} \|g(x)\|_2 \le A_q$ for some constant $A_q$, $\forall q \in \{1, 2\}$. Let $\hat{\mathcal{R}}_{w,S}$ be as defined in equation (1). $\widetilde{\mathcal{R}}(g) := \widetilde{R}_{\ell, P, \mathcal{T}}(\psi^{-1} \circ g) = \mathbb{E}_S\left[\hat{\mathcal{R}}_{w,S}(g)\right]$. Then for each $q \in \{1, 2\}$, $\forall \delta \in [0, 1]$, with probability at least $1 - \delta$,*

$$\sup_{g \in \mathcal{G}_q} \left| \hat{\mathcal{R}}_{w,S}(g) - \widetilde{\mathcal{R}}(g) \right| \le (\max_i(\left|\lambda_{\ell_{T_i}}\right| A_q + \left|\lambda_{\ell_{T_i}}\right|_0)\sqrt{2\log\frac{2}{\delta}} + CB_q \max_i\left|\lambda_{\ell_{T_i}}\right|)\sqrt{\sum_{i=1}^{N}\sum_{c=1}^{C}\frac{w_i^2}{n_{i,c}}}.$$

*where $B_q$ is a constant depending on $\mathcal{G}_q$, $\left|\lambda_{\ell_{T_i}}\right|_0 = \max_c\left|\lambda_{\ell_{T_i}}(0, c)\right|$, and $\left|\lambda_{\ell_{T_i}}\right|$ is the smallest real number such that it is a Lipschitz constant of $\lambda_{\ell_{T_i}}(\cdot, c)$ for all c.*

Theorem 11 is a special case of of Lemma 26 which extends the notion of Rademacher complexity to the LMNTM setting and applies to arbitrary function classes. Lemma 26 is presented in the appendix. Let $HM_i$ denote the harmonic mean of $n_{i,1}, \ldots, n_{i,C}$, i.e., $HM_i = \frac{C}{\sum_{c=1}^{C}\frac{1}{n_{i,c}}}$. The term $\sqrt{\sum_{i=1}^{N}\sum_{c=1}^{C}\frac{w_i^2}{n_{i,c}}}$ could be written as $\sqrt{C\sum_{i=1}^{N}\frac{w_i^2}{HM_i}}$ and is optimized by $w_i = HM_i/\sum_{m=1}^{N}HM_m$, leading to $\sqrt{\sum_{i=1}^{N}\sum_{c=1}^{C}\frac{w_i^2}{n_{i,c}}} = \sqrt{\frac{C}{\sum_{i=1}^{N}HM_i}}$. The term $\sqrt{\frac{C}{\sum_{i=1}^{N}HM_i}}$ vanishes (needed to establish consistency) when $N$ goes to infinity, or when $\exists i$ s.t. $\forall c$, $n_{i,c}$ goes to infinity. For the special case where all bags have the same size $n$ and all weights $w_i$ are $1/N$, $\sqrt{\sum_{i=1}^{N}\sum_{c=1}^{C}\frac{w_i^2}{n_{i,c}}} = \sqrt{\frac{C}{Nn}}$. Thus, consistency is possible even if bag size remains bounded. Assuming $\ell$ is the log loss and $\psi^{-1}$ is the softmax function, we next study the constants $\left|\lambda_{\ell_{T_i}}\right|$ and $\left|\lambda_{\ell_{T_i}}\right|_0$.

**Proposition 12.** *Let $\ell$ be the log loss, $\psi^{-1}$ be the softmax function, and $T$ be a column-stochastic matrix. Then $|\lambda_{\ell_T}| \le \sqrt{2}$.*

The constant $|\lambda_{\ell_T}|_0 = \max_c|\lambda_{\ell_T}(0, c)| = \max_c -\log(\frac{1}{C}\sum_{j=1}^{C}t_{c,j})$. The invertibility of $T$ guarantees $\sum_{j=1}^{C}t_{c,j}$ is positive and hence the finiteness of $|\lambda_{\ell_T}|_0$. However, if we have a "bad" $T$, $\sum_{j=1}^{C}t_{c,j}$ could be arbitrarily close to 0 leading to a large $|\lambda_{\ell_T}|_0$.

Following Theorem 11, if the function class $\mathcal{G}$ has a universal approximation property, such as an RKHS associated to a universal kernel, or an MLP with increasing number of nodes, consistency for LMNTM via (regularized) minimization of $\hat{\mathcal{R}}_{w,S}(g)$ can be shown by leveraging standard techniques, provided $N \to \infty$ (bag size may remain bounded). Then the excess risk bound in Theorem 7 would automatically imply consistency with respect to 0-1 loss.

## 5 The LLPFC algorithms

In this section, we define a probabilistic model for LLP, show how LLP reduces to LMNTM, and introduce algorithms that we refer to as the LLPFC algorithms.

### 5.1 Probabilistic Model for LLP

Given a measure $P$ on the space $\mathcal{X} \times \mathcal{Y}$, let $\{P_c : c \in \mathcal{Y}\}$ denote the class-conditional distributions of $\mathcal{X}$, i.e., $\forall$ events $\mathcal{A} \subset \mathcal{X}$, $P_c(\mathcal{A}) = P(\mathcal{A} \mid Y = c)$. Let $\sigma(c) = P(Y = c), \forall c \in \mathcal{Y}$ and call $\sigma = (\sigma(1), \ldots, \sigma(C))$ the clean prior. Assume $\forall c \in \mathcal{Y}, \sigma(c) \ne 0$. Given $z = (z(1), \ldots, z(C)) \in \Delta^C$, let $P_z$ be the probability measure on $\mathcal{X} \times \mathcal{Y}$ s.t. $\forall$ events $\mathcal{A} \subset \mathcal{X}, \forall i \in \mathcal{Y}, P_z(\mathcal{A} \times \{i\}) = z(i)P_i(\mathcal{A})$. Thus $P_z$ has the same class-conditional distributions as $P$ but a variable prior $z$.

We first define a model for a single bag. Given $z \in \Delta^C$, we say that bag $b$ is *governed* by $z \in \Delta^C$ if $b$ is a collection of feature vectors $\{X_j : j \in \mathbb{N}_{|b|}\}$ annotated by label proportion $\hat{z} = (\hat{z}(1), \hat{z}(2), \ldots, \hat{z}(C))$, where $|b|$ denotes the cardinality of the bag, each $X_j$ is paired with an unobserved label $Y_j$ s.t. $(X_j, Y_j) \overset{iid}{\sim} P_z$, and $\hat{z}(c) = \frac{1}{|b|}\sum_{j=1}^{|b|}\mathbb{1}_{Y_j=c}$. Note $\mathbb{E}_{P_z}[\hat{z}] = z$ and $P_z(Y_j = c) = z(c)$. We think of $z$ as the true label proportion and $\hat{z}$ as the empirical label proportion.

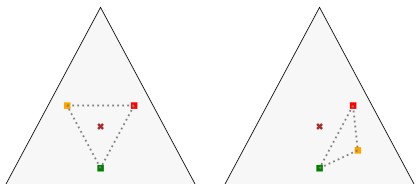

Figure 1: The gray triangle represents the probability simplex $\Delta^3$. The squares represent $\gamma_1$, $\gamma_2$, and $\gamma_3$. The cross is $\sigma$. The ternary graph on the left visualizes an example where Assumption 13 holds. The one on the right visualizes an example where Assumption 13 fails.

Using this model for individual bags, we now formally state a model for LLP. Given bags $\{b_k\}$, let each $b_k$ be governed by $\gamma_k$. Each $b_k$ is a collection of feature vectors $\left\{ X_j^k : j \in \mathbb{N}_{|b_k|} \right\}$ where $(X_j^k, Y_j^k) \overset{i.i.d.}{\sim} P_{\gamma_k}$ and $Y_j^k$ is unknown. Further assume the $X_j^k$'s are independent for all $k$ and $j$. In practice, $\gamma_k$ is unknown and we observe $\hat{\gamma}_k$ with $\hat{\gamma}_k(c) = \frac{1}{|b_k|} \sum_{j=1}^{|b_k|} \mathbb{1}_{Y_j^k = c}$ instead. The goal is learn an $f$ that minimizes the risk $\mathcal{R}_{L,P} = \mathbb{E}_{(X,Y) \sim P}[L(f(X), Y)]$ where $L$ is the 0-1 loss, given access to the training data $\{(b_k, \hat{\gamma}_k)\}$.

## 5.2 The Case of C Bags: Reduction to LLN

To explain our reduction of LLP to LLN, we first consider the case of exactly $C$ bags $b_1, b_2, \ldots, b_C$, governed by respective (unobserved) $\gamma_1, \ldots, \gamma_C \in \Delta^C$, and annotated with label proportions $\hat{\gamma}_1, \ldots \hat{\gamma}_C$. Define $\Gamma \in \mathbb{R}^{C \times C}$ by $\Gamma(i, j) = \gamma_i(j)$, and let $\Gamma^{tr}$ denote the transpose of $\Gamma$. Recall that $\sigma$ is the class prior associated to $P$. To model LLP with $C$ bags as an LLN problem, we make the following assumption on $\Gamma$ and $\sigma$:

**Assumption 13.** $\exists$ *unique* $\alpha \in \mathring{\Delta}^C$ *s.t.* $\Gamma^{tr}\alpha = \sigma$.

We write $\alpha = (\alpha(1), \ldots, \alpha(C))$. Assumption 13 is equivalent to: $\{\gamma_1, \ldots, \gamma_C\}$ is a linearly independent set and $\sigma$ is in the interior of the convex hull of $\{\gamma_1, \ldots, \gamma_C\}$. Ternary plots in Figure 5.2 visualize examples where assumption 13 holds and fails when $C = 3$. Intuitively, assumption 13 is more likely to hold when $\{\gamma_i : i \in \mathbb{N}_C\}$ are more "spread out" in $\Delta^C$, in which case it is more likely for $\sigma$ to reside in the convex hull of $\{\gamma_i : i \in \mathbb{N}_C\}$.

To reduce LLP with $C$ bags to LLN, we simply propose to assign the "noisy label" $\tilde{Y} = i$ to all elements of bag $b_i$ and to construct a noise transition matrix $T$ with $T(i, j) = \gamma_i(j)\alpha(i)/\sigma(j)$. Assumption 13 ensures $T$ is indeed a column-stochastic matrix. Thus, the probability measure $\bar{P}_T$ on $\mathcal{X} \times \mathcal{Y} \times \mathcal{Y}$ satisfies $\alpha(i) = \bar{P}_T(\tilde{Y} = i)$ and $P_{\gamma_i}(\cdot) = \bar{P}_T(\cdot \mid \tilde{Y} = i)$, which further implies $\gamma_i(c) = \bar{P}_T(Y = c \mid \tilde{Y} = i)$. We confirm these facts in Section E in the appendix. Such construction transforms LLP with $C$ bags into LLN with an estimable noise transition matrix $T$. Each element of a bag can then be viewed as a triplet $(X, Y, \tilde{Y})$, with $Y$ unobserved, such that $(X, Y)$ is drawn from $P_{\gamma_{\tilde{Y}}}$. After assigning the noisy labels, we have a dataset $\bigcup_{c=1}^C \left\{ (X_j^c, c) : j \in \mathbb{N}_{|b_c|} \right\}$ along with the noise transition matrix $T$. This allows us to leverage the forward correction loss $\ell_T$ to minimize the objective $\mathcal{R}_{\ell_T, P_T}(f) = \mathbb{E}_{P_T}[\ell_T(f(X), \tilde{Y})]$ which can be estimated by the empirical risk $\sum_{c=1}^C \frac{\alpha(c)}{|b_c|} \sum_{j=1}^{|b_c|} \ell_T(f(X_j^c), c)$.

## 5.3 The General Case: Reduction to LMNTM

More generally, consider LLP with $NC$ bags, $N \in \mathbb{N}$. We propose to randomly partition the bags into $N$ groups, each with $C$ bags indexed from 1 to $C$. Let $k_{i,c}$ denote the index of the $c$-th bag in the $i$-th group. Thus, $b_{k_{i,c}}$ is the $c$-th bag in the $i$-th group and it is governed by $\gamma_{k_{i,c}}$. For $i \in \mathbb{N}_N$, define the matrix $\Gamma_i \in \mathbb{R}^{C \times C}$ by $\Gamma_i(c_1, c_2) = \gamma_{k_{i,c_1}}(c_2), \forall c_1, c_2 \in \mathcal{Y}$. We make the following assumption on the $\Gamma_i$'s and $\sigma$:

**Assumption 14.** *For each* $i \in \mathbb{N}_N$, $\exists$ *unique* $\alpha_i \in \mathring{\Delta}^C$ *s.t.* $\Gamma_i^{tr}\alpha_i = \sigma$.

Thus, every group $i$ can be modeled as above as an LLN problem with noise transition matrix $T_i$ where $T_i(c_1, c_2) = \gamma_{k_{i,c_1}}(c_2)\alpha_i(c_1)/\sigma(c_2)$. Data points in the bag assigned with noisy label $c$ in the $i$-th group can be viewed as drawn $i.i.d.$ from the class conditional distribution $P_{T_i}(\cdot \mid c)$. This

---

**Algorithm 1** LLPFC-ideal

1: **Input:** $\{(b_k, \gamma_k)\}_{k=1}^{NC}$ and $w \in \Delta^N$ where $b_k = \{X_j^k : j \in \mathbb{N}_{|b_k|}\}$.
2: Randomly partition the bags into $N$ groups $\{G_i\}_{i=1}^N$ s.t. $G_i = \{(b_{k_{i,c}}, \gamma_{k_{i,c}}) : c \in \mathcal{Y}\}$
3: **for** $i = 1 : N$ **do**
4:     $\Gamma_i \leftarrow [\gamma_{k_{i,1}}, \gamma_{k_{i,2}}, \dots, \gamma_{k_{i,C}}]^{tr}$
5:     $\alpha_i \leftarrow \Gamma_i^{-tr} \sigma$
6:     **for** $c_1 = 1 : C, c_2 = 1 : C$ **do**
7:        $T_i(c_1, c_2) \leftarrow \gamma_{k_{i,c_1}}(c_2) \alpha_i(c_1) / \sigma(c_2)$
8:     **end for**
9: **end for**
10: Train $f$ with the empirical objective (1)

---

**Algorithm 2** LLPFC-uniform

1: **Input:** $\{(b_k, \hat{\gamma}_k)\}_{k=1}^{NC}$ and $w \in \Delta^N$ where $b_k = \{X_j^k : j \in \mathbb{N}_{|b_k|}\}$.
2: Partition the bags as step 2 in Algorithm 1.
3: **for** $i = 1 : N$ **do**
4:     $\hat{\Gamma}_i \leftarrow [\hat{\gamma}_{k_{i,1}}, \hat{\gamma}_{k_{i,2}}, \dots, \hat{\gamma}_{k_{i,C}}]^{tr}$
5:     $n_i \leftarrow \sum_{c=1}^C |b_{k_{i,c}}|$
6:     $\hat{\alpha}_i(c) \leftarrow |b_{k_{i,c}}|/n_i$ for each $c = 1 : C$
7:     $\hat{\sigma}_i \leftarrow \hat{\Gamma}_i^{tr} \hat{\alpha}_i$
8:     **for** $c_1 = 1 : C, c_2 = 1 : C$ **do**
9:        $\hat{T}_i(c_1, c_2) \leftarrow \hat{\gamma}_{k_{i,c_1}}(c_2) \hat{\alpha}_i(c_1) / \hat{\sigma}_i(c_2)$
10:     **end for**
11: **end for**
12: Train with $\sum_{i,c} \frac{w_i}{n_i} \sum_j \ell_{\hat{T}_i}(f(X_j^{k_{i,c}}), c)$.

---

problem now maps directly to LMNTM as described in Section 4.1, and satisfies the associated performance guarantees. In the next subsection, we spell out the associated algorithm.

## 5.4 Algorithms

As above, assume we have $NC$ bags where $N \in \mathbb{N}$. Let each bag $b_k$ be governed by $\gamma_k \in \Delta^C$ and be annotated by label proportion $\hat{\gamma}_k$. We first present the LLPFC-ideal algorithm in an ideal setting where $\sigma$, the $\gamma_k$'s and the $\alpha_i$'s are known precisely and Assumption 14 holds. We then present the real-world adaptations LLPFC-uniform and LLPFC-approx in practical settings.

The LLPFC-ideal algorithm is presented in Algorithm 1. We follow the idea in section 5.3 to partition the bags into $N$ groups of $C$ bags, and model each group as an LLN problem. In Algorithm 1, we assume $\gamma_k$ and $\sigma$ are known and Assumption 14 holds. The theoretical analysis in Section 4 is immediately applicable to the LLPFC-ideal algorithm. We partition the bags by uniformly randomly partitioning the set of indices $\mathbb{N}_{NC}$ into disjoint subsets $\{k_{i,c} : c \in \mathcal{Y}\}$, $i \in \mathbb{N}_N$, where $k_{i,c}$ denotes the index of the $c$-th bag in the $i$-th group. We denote the inverse transpose of $\Gamma_i$ by $\Gamma_i^{-tr}$.

In practice, when $\gamma_k$ is unknown, we replace $\gamma_k$ with $\hat{\gamma}_k$ as a plug-in method. Hence, we work with $\hat{\sigma} = \frac{\sum_{k=1}^{NC} |b_k| \hat{\gamma}_k}{\sum_{k=1}^{NC} |b_k|}$ and $\hat{\Gamma}_i = [\hat{\gamma}_{k_{i,1}}, \hat{\gamma}_{k_{i,2}}, \dots, \hat{\gamma}_{k_{i,C}}]^{tr}$ instead of $\sigma$ and $\Gamma_i$ in Algorithm 1, respectively. Here $\hat{\sigma}$ is the label proportion of all training data points and we use it as an estimate of the clean prior $\sigma$. Likewise, $\alpha_i = \Gamma_i^{-tr} \sigma$ in Algorithm 1 should be replaced with $\hat{\alpha}_i = \hat{\Gamma}_i^{-tr} \hat{\sigma}$ and we would like to use $\hat{\Gamma}_i$, $\hat{\sigma}$, and $\hat{\alpha}_i$ to calculate $\hat{T}_i$ as an estimate of $T_i$. For this to make sense, we need $\hat{\alpha}_i = \hat{\Gamma}_i^{-tr} \hat{\sigma} \in \mathring{\Delta}^C$, which is equivalent to $\hat{\sigma}$ being in the interior of the convex hull of $\{\hat{\gamma}_{k_{i,c}} : c \in \mathcal{Y}\}$ for all $i$. However, this may not be the case in practice. Thus, we consider two

---

**Algorithm 3** LLPFC-approx

1: **Input:** $\{(b_k, \hat{\gamma}_k)\}_{k=1}^{NC}$ and $w \in \Delta^N$ where $b_k = \{X_j^k : j \in \mathbb{N}_{|b_k|}\}$.
2: $\hat{\sigma} \leftarrow \frac{\sum_{k=1}^{NC} |b_k| \hat{\gamma}_k}{\sum_{k=1}^{NC} |b_k|}$
3: Partition the bags as step 2 in Algorithm 1.
4: **for** $i = 1 : N$ **do**
5:     $\hat{\Gamma}_i \leftarrow [\hat{\gamma}_{k_{i,1}}, \hat{\gamma}_{k_{i,2}}, \dots, \hat{\gamma}_{k_{i,C}}]^{tr}$
6:     $\hat{\alpha}_i \leftarrow \arg\min_{\alpha \in \Delta^C} ||\hat{\sigma} - \hat{\Gamma}_i^{tr} \alpha||_2^2$
7:     $\hat{\sigma}_i \leftarrow \hat{\Gamma}_i^{tr} \hat{\alpha}_i$
8:     **for** $c_1 = 1 : C, c_2 = 1 : C$ **do**
9:        $\hat{T}_i(c_1, c_2) \leftarrow \hat{\gamma}_{k_{i,c_1}}(c_2) \hat{\alpha}_i(c_1) / \hat{\sigma}_i(c_2)$
10:     **end for**
11: **end for**
12: Train with $\sum_{i,c} \frac{w_i \hat{\alpha}_i(c)}{|b_{k_{i,c}}|} \sum_j \ell_{\hat{T}_i}(f(X_j^{k_{i,c}}), c)$

---

heuristics to estimate $\hat{T}_i$ as real-world adaptations of the LLPFC-ideal algorithm. The first, called LLPFC-uniform, is presented in Algorithm 2 which sets $\hat{\alpha}_i$ by counting the occurrences of the noisy labels. This is motivated by our model wherein $\alpha_i$ is the noisy class prior for the $i$-th group. The second, called LLPFC-approx, is presented in Algorithm 3 and sets $\hat{\alpha}_i$ to be the solution of $\arg\min_{\alpha \in \Delta^C} ||\hat{\sigma} - \hat{\Gamma}_i \alpha||_2^2$. It should be noted that in both practical algorithms, we use a different $\hat{\sigma}_i$ as an estimate of $\sigma$ for each group, to ensure that each $\hat{T}_i$ is a column-stochastic matrix. In experiments where we have $NC + k$ number of bags with $0 < k < C$, we can randomly resample $NC$ number of bags and regroup them in every few epochs. Both real-world adaptations perform reasonably well in experiments.

## 6  Experiments

[1] We compare against three previous works that have studied LLP applying deep learning to image

Table 1: Test Accuracy for Wide ResNet-16-4

| Data set | Method | 32 | 64 | 128 | 256 | 512 | 1024 | 2048 |
|---|---|---|---|---|---|---|---|---|
| CIFAR10 | KL | .4255 ± .13 | .6817 ± .16 | .5346 ± .11 | .3749 ± .14 | .2938 ± .04 | Out of RAM | Out of RAM |
| | LLPVAT | .4911 ± .15 | .5137 ± .22 | .4744 ± .12 | .4423 ± .16 | Out of RAM | Out of RAM | Out of RAM |
| | LLPFC-uniform | .7926 ± .01 | **.7683 ± .02** | .7399 ± .02 | .7381 ± .01 | .7224 ± .01 | .7182 ± .01 | .6925 ± .03 |
| | LLPFC-approx | **.7993 ± .00** | .7671 ± .01 | **.7528 ± .01** | **.7404 ± .00** | **.7409 ± .02** | **.7205 ± .03** | **.7283 ± .02** |
| SVHN | KL | .2465 ± .10 | .1152 ± .07 | .1022 ± .03 | .1294 ± .04 | .1039 ± .04 | Out of RAM | Out of RAM |
| | LLPVAT | .2675 ± .36 | .1398 ± .08 | .1004 ± .03 | .1294 ± .04 | Out of RAM | Out of RAM | Out of RAM |
| | LLPFC-uniform | **.9012 ± .02** | **.8855 ± .02** | .8760 ± .02 | .8736 ± .01 | .8681 ± .02 | **.8709 ± .02** | .8717 ± .01 |
| | LLPFC-approx | .8903 ± .02 | .8844 ± .02 | **.8815 ± .03** | **.8808 ± .01** | **.8771 ± .02** | .8701 ± .02 | **.8738 ± .01** |
| EMNIST | KL | .8413 ± .04 | .8637 ± .04 | .9111 ± .00 | .5361 ± .12 | .0845 ± .01 | .0826 ± .01 | Out of RAM |
| | LLPVAT | .8254 ± .04 | .9045 ± .02 | **.9136 ± .00** | .5071 ± .09 | .0859 ± .01 | Out of RAM | Out of RAM |
| | LLPFC-uniform | **.9165 ± .01** | .9061 ± .01 | .9015 ± .01 | .8790 ± .03 | .8886 ± .02 | .8461 ± .05 | .8817 ± .01 |
| | LLPFC-approx | .9092 ± .01 | **.9074 ± .01** | .9065 ± .00 | **.8993 ± .01** | **.9048 ± .00** | **.8969 ± .01** | **.9007 ± .01** |

Table 2: Test Accuracy for ResNet18

| Data set | Method | 32 | 64 | 128 | 256 | 512 | 1024 | 2048 |
|---|---|---|---|---|---|---|---|---|
| CIFAR10 | KL | .7837 ± .01 | **.7565 ± .01** | .6918 ± .01 | .6106 ± .04 | .5696 ± .05 | .5197 ± .04 | .4576 ± .03 |
| | LLPVAT | **.7907 ± .01** | .7499 ± .01 | **.6946 ± .01** | **.6115 ± .03** | .5670 ± .04 | .4881 ± .02 | .4624 ± .02 |
| | LLPFC-uniform | .6601 ± .01 | .6310 ± .01 | .5867 ± .01 | .5603 ± .01 | .5670 ± .01 | .5623 ± .01 | .5288 ± .03 |
| | LLPFC-approx | .6567 ± .01 | .6136 ± .01 | .5997 ± .01 | .5931 ± .02 | **.6062 ± .01** | **.6169 ± .01** | **.5591 ± .04** |
| SVHN | KL | .1279 ± .06 | .0716 ± .01 | .3042 ± .30 | .1026 ± .04 | .2489 ± .24 | .3123 ± .29 | .2797 ± .12 |
| | LLPVAT | .1279 ± .06 | .1939 ± .27 | .6962 ± .34 | .3673 ± .39 | .4003 ± .31 | .3999 ± .33 | .3736 ± .24 |
| | LLPFC-uniform | .8823 ± .01 | .8644 ± .01 | .8433 ± .01 | .8390 ± .01 | .8360 ± .00 | .8086 ± .02 | .8188 ± .01 |
| | LLPFC-approx | **.8824 ± .01** | **.8672 ± .01** | **.8570 ± .01** | **.8483 ± .01** | **.8492 ± .01** | **.8498 ± .01** | **.8534 ± .01** |
| EMNIST | KL | **.9319 ± .00** | .9295 ± .00 | **.9306 ± .00** | .9269 ± .00 | **.9267 ± .00** | **.9239 ± .00** | .9106 ± .01 |
| | LLPVAT | .9308 ± .00 | **.9299 ± .00** | .9299 ± .00 | **.9281 ± .00** | .9248 ± .00 | .9222 ± .00 | **.9128 ± .00** |
| | LLPFC-uniform | .9144 ± .00 | .8954 ± .00 | .8744 ± .00 | .8600 ± .00 | .8448 ± .00 | .8388 ± .01 | .8245 ± .01 |
| | LLPFC-approx | .9146 ± .00 | .8998 ± .00 | .8874 ± .00 | .8764 ± .01 | .8670 ± .00 | .8736 ± .01 | .8660 ± .01 |

Table 3: Test Accuracy for VGG16

| Data set | Method | 32 | 64 | 128 | 256 | 512 | 1024 | 2048 |
|---|---|---|---|---|---|---|---|---|
| CIFAR10 | KL | .2513 ± .11 | .2130 ± .06 | .1794 ± .04 | .1160 ± .02 | .1117 ± .01 | .1221 ± .00 | .1049 ± .01 |
| | LLPVAT | .4634 ± .07 | .2093 ± .03 | .1399 ± .03 | .1145 ± .02 | .1172 ± .02 | .1189 ± .00 | Out of RAM |
| | LLPFC-uniform | **.7602 ± .00** | **.7372 ± .01** | **.7300 ± .01** | **.7226 ± .01** | **.7136 ± .01** | **.7111 ± .01** | **.7033 ± .03** |
| | LLPFC-approx | .7566 ± .01 | .7310 ± .01 | .7003 ± .01 | .7004 ± .01 | .6870 ± .01 | .6857 ± .02 | .6645 ± .03 |
| SVHN | KL | .1277 ± .06 | .0893 ± .04 | .1054 ± .05 | .1024 ± .05 | .1104 ± .04 | .0885 ± .01 | .1372 ± .03 |
| | LLPVAT | .1117 ± .05 | .0736 ± .01 | .1051 ± .05 | .1023 ± .06 | .1125 ± .04 | .1061 ± .05 | Out of RAM |
| | LLPFC-uniform | .4177 ± .15 | .4708 ± .23 | **.5402 ± .21** | **.1734 ± .11** | **.4249 ± .30** | **.5691 ± .27** | **.6869 ± .13** |
| | LLPFC-approx | **.4299 ± .28** | **.4994 ± .23** | .1091 ± .04 | .1188 ± .05 | .1903 ± .14 | .4097 ± .17 | .4429 ± .18 |
| EMNIST | KL | .5952 ± .45 | .2348 ± .22 | .0974 ± .01 | .0842 ± .02 | .0702 ± .01 | .0692 ± .01 | .0597 ± .02 |
| | LLPVAT | .8593 ± .16 | .3329 ± .33 | .1042 ± .01 | .0833 ± .02 | .0696 ± .01 | .0711 ± .00 | Out of RAM |
| | LLPFC-uniform | **.9311 ± .00** | .9279 ± .00 | **.9258 ± .00** | **.9242 ± .00** | **.9239 ± .00** | **.9233 ± .00** | **.9220 ± .00** |
| | LLPFC-approx | .9310 ± .00 | **.9280 ± .00** | .9249 ± .00 | .9240 ± .00 | .9227 ± .00 | .9206 ± .00 | .9205 ± .00 |

Table 4: Test Accuracy for LLPGAN architecture

| Data set | Method | 32 | 64 | 128 | 256 | 512 | 1024 | 2048 |
|---|---|---|---|---|---|---|---|---|
| CIFAR10 | LLPGAN | .3630 ± .01 | .3133 ± .02 | .3328 ± .03 | .3363 ± .03 | .3460 ± .03 | .2824 ± .05 | .2236 ± .08 |
| | LLPFC-uniform | .6145 ± .01 | .5826 ± .01 | .5565 ± .03 | .5452 ± .01 | .5511 ± .02 | .5358 ± .01 | .5438 ± .03 |
| | LLPFC-approx | **.6169 ± .01** | **.5875 ± .01** | **.5642 ± .02** | **.5687 ± .02** | **.5621 ± .03** | **.5610 ± .01** | **.5567 ± .02** |
| SVHN | LLPGAN | .2378 ± .24 | .7135 ± .06 | .7680 ± .04 | .6058 ± .29 | .4863 ± .22 | .1725 ± .06 | .1382 ± .04 |
| | LLPFC-uniform | **.8800 ± .00** | **.8581 ± .01** | **.8480 ± .01** | .8393 ± .01 | .8347 ± .01 | .8258 ± .01 | .8327 ± .01 |
| | LLPFC-approx | .8779 ± .01 | .8519 ± .01 | .7061 ± .33 | **.8453 ± .02** | **.8423 ± .01** | **.8386 ± .01** | **.8527 ± .01** |

data: Dulac-Arnold et al. [9] study the KL loss described in the introduction, and a novel loss

---

[1]Code is available at https://github.com/Z-Jianxin/LLPFC

based on optimal transport. They find that KL performs just as well as the novel loss. Liu et al. [22] employ the KL loss within a generative adversarial framework (LLPGAN). Tsai and Lin [46] propose augmenting the KL loss with a regularizer from semi-supervised learning and show improved performance (LLPVAT). We compare both LLPFC-uniform and LLPFC-approx against the KL loss, LLPGAN, and LLPVAT to clearly establish which empirical objective is better. Recent papers on multiclass LLP for which code is not available were not included [23, 16].

We generate bags with fixed, equal sizes in $\{32, 64, 128, 256, 512, 1024, 2048\}$. To generate each bag, we first sample a label proportion $\gamma$ from the uniform distribution on $\Delta^C$. Then we sample data points from a benchmark dataset without replacement using a multinomial distribution with parameter $\gamma$. It should be noted that Tsai and Lin [46], Dulac-Arnold et al. [9], and Liu et al. [22] generate bags by shuffling all data points and making every $B$ data points a bag where $B$ is a fixed bag size. Their method is equivalent to sampling data points without replacement using a multinomial distribution with a fixed parameter $\gamma = \left\{\frac{1}{C}, \frac{1}{C}, \ldots, \frac{1}{C}\right\}$. As noted by Scott and Zhang [39], this leads to bags with very similar label proportions which makes the learning task much more challenging.

We repeat each experiment 5 times and report the mean test accuracy and standard deviation. All models are trained on a single Nvidia Tesla v100 GPU with 16GB RAM. In our implementation of LLPFC algorithms, the weight $w$ is set to be $(\frac{1}{N}, \ldots, \frac{1}{N}) \in \Delta^N$ and our choice of the proper composite loss is the cross-entropy loss.

For the comparison against KL and LLPVAT, we perform experiments on three benchmark image datasets: the "letter" split of EMNIST [6], SVHN [29], and CIFAR10 [17]. To show that our approach is robust to the choice of architecture, we experiment with three different networks: Wide ResNet-16-4 [52], ResNet18 [11], and VGG16 [42]. We train these networks with the parameters suggested in the original papers. The test accuracies are reported in Tables 1, 2, and 3. Since convergence in the GAN framework is sensitive to the choice of architecture and hyperparameters, we compare LLPFC against LLPGAN using the architecture proposed in the original paper along with the hyperparameters suggested in their code[2]. It should be noted that for LLPFC we only use the discriminator for classification and did not use the generator to augment data. Since Liu et al. [22] only provide hyperparameters for colored images, we perform experiments on SVHN and CIFAR10 only. The test accuracies are reported in Table 4.

LLPFC-uniform and LLPFC-approx substantially outperform the competitors in a clear majority of settings. The experiment results clearly establish our methods as the state-of-the-art by a substantial margin. All three competitors perform gradient descent with minibatches of bags and the GPU at times runs out of memory when the bag size is large. Our implementation, which also uses stochastic optimization, does not suffer from this phenomenon. Full experimental details are in the appendix.

## 7 Conclusions and Future Work

We propose a theoretically supported approach to LLP by reducing it to learning with label noise and using the forward correction (FC) loss. An excess risk bound and generalization error analysis are established. Our approach outperforms leading existing methods in deep learning scenarios across multiple datasets and architectures. A limitation of our approach is that the theory makes an assumption that may not be verifiable in practice. Future research directions include optimizing the grouping of bags and adapting LLPFC to other objectives beyond accuracy.

**Acknowledgement** The authors were supported in part by the National Science Foundation under awards 1838179 and 2008074, and by the Department of Defense, Defense Threat Reduction Agency under award HDTRA1-20-2-0002.

---

[2]https://github.com/liujiabin008/LLP-GAN

