# OpenReview forum: "Learning from Label Proportions by Learning with Label Noise"
_NeurIPS.cc/2022/Conference — NeurIPS 2022 Accept_

### Official Review · Reviewer_djWm · 2022-07-11

**Rating:** 7
**Confidence:** 5
**Soundness:** 4 excellent
**Presentation:** 2 fair
**Contribution:** 4 excellent

**Summary:**

Multi-class label proportion learning (LLP) is a weakly supervised learning problem that aims at obtaining a multi-class classifier with only bags (each bag is a collection of data points) and the proportions of different classes contained in the bags. In this paper, it is shown that the problem of label proportion learning can be solved efficiently with the techniques in the field of noisy label learning (LLN). The authors first give the consistency results on the Forward Correction (FC) technique, which is one of the prevailing methods in the field of  LLN, and then extend it to enable the use of multiple transition matrices. Under some mild assumptions, the authors show that the problem of LLP can be solved with the proposed extension of FC. Experiments on benchmark datasets are conducted to show the efficiency of the proposed method.

**Questions:**

There can be quite a number of assignments of labels w.r.t. different bags and thus the construction of transition matrices can be flexible. For example, in the scenario mentioned in Section 5.2, there can be C! different assignments. If there exists an optimal assignment of pseudo noisy labels?

**Limitations:**

The authors have addressed the limitation of assumptions made in Section 5.1.

**Strengths And Weaknesses:**

Strengths:

1. A novel extension of FC technique is proposed that removes the limitation on the number of transition matrices.

2. A clear connection between LLP and LLN is constructed with a strategy of converting label proportion to noisy labels and noise-transition matrices. With this conversion, the problem of multi-class LLP can be solved with consistency guarantee.

3. Rigorous theoretical analyses are conducted to support all the claims made in this paper. The provided generalization error bound well depicts factors that have influence on the proposed method.

Weaknesses:

The presentation of this paper should be improved for better readability. LLPFC, the main result of this paper, appears in the latter part of this paper, which makes the key point of this paper ambiguous. I think this problem can be mitigated by exchanging the order of Section 4.3 and Section 5.

---

> ### Author Response · Authors · 2022-08-02
> **Author Response**
>
> Thank you for your advice and questions.
>
> ## Ordering of sections:
> We experimented with different organizations of the sections as you suggest, but ultimately found that the order in the original submission made the most sense. We did our best to keep the reader's attention focused on the goal of LLP.
>
> ## Optimal Construction of Noise Transition Matrices:
> To optimize the weights or the assignment of bags we would need to optimize the composition of our two bounds:
> $\theta ( R_{L, P} (f)  - R_{L, P}^* ) \leq Empirical Risk + Generalization Error Bound - R^*_{l, P, T}$. This is in contrast to the approach with backward correction [Scott and Zhang 2020] which does not require the excess risk bound (because their excess target risk is simply proportional to the excess surrogate risk). Therefore, to optimize the composition of our bounds, we'd need to estimate the surrogate Bayes risk, a challenging task. We also note that both the generalization error bound and excess risk bound involves weights $w_i$ and noise matrices $T_i$. Therefore, even if the surrogate Bayes risk were somehow known, the resulting integer programming problem is much more involved than for the backward correction, where it's a simple matching problem. For example, with the cross-entropy loss and using our bound in Proposition 6 and 12 in the original submission (corresponding to proposition 11 and 22 in the revision, respectively), we would need to optimize for the following bound:
>
> $ R_{L, P}(f) -  R_{L, P}^* \leq \sqrt{\frac{1}{\sum_{i} \frac{w_i}{2||T_i^{-1}||^2}}} \sqrt{Empirical Risk - {R}^*_{l, P, {T}} + (A  + B - D \min_{i, c} \log (\frac{1}{C} \sum_{j=1}^C T_i(c, j)))\sqrt{\sum_{i,c}\frac{w_i^2}{n_{i,c}}}} $
> where $A$, $B$, and $D$ are positive constants depending on the function space, the number of classes $C$, and $\delta$. $T_i(c,j)$ is the entry at $c$-th row and $j$-th column in the matrix $T_i$.
>
> Fortunately, LLPFC with random partitioning and uniform weights demonstrates superior empirical results in the experiments and outperforms other multiclass LLP methods by a significant margin. We believe weight optimization is much more important for the backward correction, where the loss functions can have large and disparate magnitudes (which need to be offset by carefully chosen weights), than it is for forward correction where the outputs of the inverse link function are in the unit simplex and thus all of a comparable magnitude. A similar point is made by [Patrini et al. 2017](https://openaccess.thecvf.com/content_cvpr_2017/papers/Patrini_Making_Deep_Neural_CVPR_2017_paper.pdf) in the last two sentences in the first paragraph of section 6.

---

### Official Review · Reviewer_SBPr · 2022-07-11

**Rating:** 5
**Confidence:** 2
**Soundness:** 3 good
**Presentation:** 2 fair
**Contribution:** 3 good

**Summary:**

The paper focused on learning from label proportions (LLP) task. First, they show that background theory of forward correction loss in learning with label noise (LLN), and extend the results with multiple noise transition matrices. Next, they propose LLPFC algorithm with probabilistic model for reducing LLP to LLN with multiple noise transition matrices.

**Questions:**

- It would be nice to provide the explanation of the experiment results and what they support.
- Is it easy to estimate the clean prior $\sigma(c)$?
- Are Assumption 13 and 14 empirically reasonable in LLP?


**Limitations:**

They addressed the limitations in the paper.

**Strengths And Weaknesses:**

### Strengths

- The connection between LLP and LLN is reasonable and novel.
- They provide theoretical analysis and this would be useful for both LLN and LLP community.

### Weaknesses

- Lack of the experiment results
  - The paper did not provide any empirical results in the main paper.
  - Also, there are no/few explanation of the main experiment results in both main and supplementary paper.
- Gap of theory and practice
  - The theoretical analysis has many assumptions, but they did not give theoretical or empirical explanations to show that the assumption is reasonable in practice.

---

> ### Author Response · Authors · 2022-08-02
> **Author Response**
>
> Thank you for comments. We have addressed these concerns in a revised version of the paper, and offer further explanation below.
>
> ## Experiment Results:
>
> We did in fact summarize our experimental findings in the last paragraph of section 6 in the original submission. We felt no additional discussion was needed given the excellent performance of our method, but in the revision, we have added a few comments to nuance the discussion. We've also moved some remarks on the theoretical results to the appendix so we can fit tables of experimental results in the main paper. Given that one extra page is allowed in the camera-ready version, we can move those remarks back to the main paper if this paper is accepted.
>
> ## Assumptions:
>
> Assumption 14 is a generalization of Assumption 13 (These are now assumptions 10 and 9 in the revision, respectively). Both of them are unverifiable in the real-world as it requires us to know the probability distribution governing the data. However, we have developed practical heuristics in the LLPFC-uniform and LLPFC-approx algorithms for estimating the noise transition parameters that do not require Assumption 14 to be true. The strength of our experimental results supports the validity of the approach. As to whether Assumptions 13 and 14 are true in practice, we offered a comment on this in the submission in the paragraph after Assumption 13, and in Fig. 1.

---

> > ### Comment · Reviewer_SBPr · 2022-08-09
> > **Thanks for your response**
> >
> > I would like to thank the authors for the answers. After I read the author responses and other reviews, I raise my score to "Borderline reject" to "Borderline accept" since most of my concerns have been addressed. However, as the other reviewers also commented, the presentation of the paper should be improved such as complex notations, in-line equations, tables of experiment section.

---

### Official Review · Reviewer_tEEw · 2022-07-13

**Rating:** 5
**Confidence:** 4
**Soundness:** 3 good
**Presentation:** 2 fair
**Contribution:** 3 good

**Summary:**

-This paper considers a conventional weakly supervised learning problem called learning from label proportions, where the training data are given in the form of bags and the proportions of each class within each bag are also given; the goal is to train a classifier predicting instance labels from such bag data. To solve the problem, they partition the bags into groups and model each group as a standard label noise problem. Next, they apply the forward correction loss, which was proposed for dealing with class-conditional label noise, to each group and study the generalization error bound of the proposed method. They also conduct some experiments to verify the effectiveness of the proposed method on some benchmark datasets.


**Questions:**

-The proposed method randomly partitions the given bags into groups and then formulates the problem of learning with multiple noise transition matrices. This random partition step seems problematic to me. As different partitions may lead to different noise transition matrices, which determine the difficulty of each LLN problem and will significantly affect the performance of the learned classifier. In Scott and Zhang 2020, they have discussed that the optimal pairing of bags can be obtained by solving an integer programming problem. In the current paper, could the authors provide similar analysis on the optimal partition strategy?

-In theorem 7, the weights w_i are expected to reweight noiser or larger subsets of data differently. But it is still unclear to me how to choose them in practice.

-In Section 5.3, the authors discuss a general case of LLP with NC bags. What if we are given m bags and m is not divisible by C? How to deal with such a general case in the current learning framework?


**Limitations:**

-The paper has some unclear parts that need to be carefully modified, and the organization of the paper can be improved.

**Strengths And Weaknesses:**

pros:

-The paper studies the problem of learning from label proportions in a theoretically grounded way. They provide calibration analysis of the forward correction loss and then apply it to their problem setting. They also derive the generalization error bound of the proposed method under some assumptions.

cons:

-The title of the paper is a bit misleading, as the problem settings of learning from label proportions (LLP) and learning with label noise (LLN) are not mutually exclusive. To be specific, both of them can be seen as learning from corrupted distributions, where the class-conditional densities p(x/y) are corrupted in the former setting (LLP), while the class posterior probabilities p(y/x) are corrupted in the latter setting (LLN). In the binary case, the former is also called mutually contaminated (MC) learning, and the latter is also called class-conditional label noise (CCN) learning. It is proved that CCN is a special case of MC, see sections 2.2-2.3 in the following paper for details:

Menon et al. "Learning from corrupted binary labels via class-probability estimation." International conference on machine learning. PMLR, 2015.

Therefore, my feeling is that, in a general viewpoint, LLP can also be seen as a kind of label noise problem, and the title may cause some confusion that how to solve one label noise problem by another label noise problem.


-The writing of the paper is unclear and some parts are hard to follow. For example, in lines 40-42, the authors claim that Patrini et al. 2017 proposed forward correction loss to remedy the backward correction loss since it is poorly suited for multi-class deep learning settings; however, in the original Patrini et al. 2017 paper, they seem only mention that forward correction is an alternative method that corrects the model predictions (see Section 4.2 in Patrini et al. 2017 paper), without any discussions on how it remedies the backward correction. Also in lines 50-52, it seems that the mutual contamination model is not that different from the noise transition matrix used in this paper, which can be seen as a 2 by 2 noise matrix corrupting the class-conditionals.

-Some sections seem incomplete. Especially for the algorithm section, how to estimate the noise matrix is an important step for the proposed method, but the details of these steps are not included in the main paper. And there are no experimental results provided in the main paper. I strongly recommend simplifying section 4.3, as the definitions are mainly borrowed from existing papers and may be moved to the appendix, and using the saved space for discussing the noise matrix estimation algorithm and reporting experimental results.

---

> ### Author Response · Authors · 2022-08-02
> **Author Response**
>
> Thank you for the detailed review and the informative advice. We believe that all of your concerns have been addressed by minor revisions, which are now reflected in the revised submission (or will be added to the camera-ready version) as described below.
>
> ## Title:
> We do not argue that LLP and LLN are mutually exclusive. If the reviewers/AC prefer, and if the system allows it, we would be willing to change the title to something like ``Learning from Label Proportions with the Forward Correction Loss" to address your concerns.
>
> ## Patrini's comments on forward/backward correction:
> Thank you for catching this. Indeed, Patrini et al. don't comment on how forward correction remedies backward correction in their original paper. Our point is that Patrini et al. 2017 suggest that forward correction outperforms backward correction in multiclass deep learning context. In [Patrini et al. 2017, CVPR], they comment that "backward correction
> does not seem to work well in the low noise regime" (the second last sentence in section 5.1.) and "backward correction
> is a linear combination of losses for every possible label,
> with coefficients that can be far by orders of magnitude and
> thus makes the learning harder" (the second last sentence in the first paragraph of section 6). We've reworded this part on line 41-46 in the revision to convey our point more clearly.
>
> ## MC and FC loss:
> We agree that the mutual contamination framework and noise transition framework are not all that different. However, our proof that FC loss is calibrated with respect to the 0-1 loss (Theorem 5) requires a class conditional noise model bound and in particular proper composite losses which are not at play with the backward correction loss and mutual contamination framework. This additional aspect of the analysis is one way in which we go beyond ideas and techniques in Scott and Zhang. The revision at line 47-57 now better expresses this point.
>
> ## Incomplete sections:
> Thanks for suggesting that we move Defs 9 and 10 to the appendix. We have done so in the revision and used the space to further discuss our algorithms and experiments. We've also moved some remarks on the theoretical results to appendix so we can fit tables of experimental results in the main paper. Given that one extra page is allowed in the camera-ready version, we can move the those remarks back to the main paper if this paper is accepted.
>
> ## Random Partitioning of Bags:
> To optimize the weights or the assignment of bags we would need to optimize the composition of our two bounds: $\theta (R_{L, P}(f)  - R_{L, P}^*) \leq Empirical Risk + Generalization Error Bound - R^*_{l, P, T}$. This is in contrast to the approach with backward correction [Scott and Zhang 2020] which does not require the excess risk bound (because their excess target risk is simply proportional to the excess surrogate risk). Therefore, to optimize the composition of our bounds, we'd need to estimate the surrogate Bayes risk, a challenging task. We also note that both the generalization error bound and excess risk bound involves weights $w_i$ and noise matrices $T_i$. Therefore, even if the surrogate Bayes risk were somehow known, the resulting integer programming problem is much more involved than for the backward correction, where it's a simple matching problem. For example, with the cross-entropy loss and using our bound in Proposition 6 and 12 in the original submission (corresponding to proposition 11 and 22 in the revision, respectively), we would need to optimize for the following bound:
> $R_{L, P}(f)-R_{L, P}^* \leq \sqrt{\frac{1}{\sum_{i} \frac{w_i}{2||T_i^{-1}||^2}}} \sqrt{Empirical Risk - R^*_{l, P, T} + (A  + B - D \min_{i, c} \log (\frac{1}{C} \sum_{j=1}^C T_i(c, j)))\sqrt{\sum_{i,c}\frac{w_i^2}{n_{i,c}}}}$
> where $A$, $B$, and $D$ are positive constants depending on the function space, the number of classes $C$, and $\delta$. $T_i(c,j)$ is the entry at $c$-th row and $j$-th column in the matrix $T_i$.
>
> Fortunately, LLPFC with random partitioning and uniform weights yields superior empirical results in the experiments and outperforms other multiclass LLP methods by a significant margin. We believe weight optimization is much more important for the backward correction, where the loss functions can have large and disparate magnitudes (which need to be offset by carefully chosen weights), than it is for forward correction where the outputs of the inverse link function are in the unit simplex and thus all of a comparable magnitude. A similar point is made by [Patrini et al. 2017, CVPR] in the last two sentences in the first paragraph of section 6.
>
> ## Number of bags:
> Thanks for pointing this out. Say we have $NC+k$ number of bags with $0< k <C$. In our experiments, we randomly resample $NC$ bags every few epochs and regroup them, which addresses this issue. We've added this discussion to section 5.4 in the revision.

---

> > ### Comment · Reviewer_tEEw · 2022-08-10
> > **Thanks for the authors' response!**
> >
> > Thanks for the answers to my questions. It is interesting to know that weight optimization is much more important for backward correction than it is for forward correction. I also read other reviews and the revised paper. The presentation has been improved a lot with the modified algorithms and experiments sections. I like your proposed new title ``Learning from Label Proportions with the Forward Correction Loss", which gets straight to the point and avoids misunderstandings. I would recommend the authors use this title instead.

---

> ### Comment · Reviewer_vSZS · 2022-08-09
> **Noise Matrix Estimation**
>
> You review seems to ask as to how the noise matrix is estimated. Unless I am misunderstanding something, the key point is that there is no need to estimate the noise matrix -- since you know the actual label priors (from the bags) and the observed label priors (from the selected label assignment), you can just set the noise matrix to contain transitions from the assigned label to the observed labels with the rates of the labels in the bags. Though I guess this solution is not unique, so a clarifying comment in the paper would be helpful.

---

> > ### Author Response · Authors · 2022-08-09
> > **Noise Matrix Estimation**
> >
> > Thanks for the clarification from vSZS. Indeed, our theory does not require a specific estimation algorithm of $T$. If Assumption 10 holds, the exact $T$ can be computed using the priors and label proportions. The quantities we used to calculate $T$, the priors and label proportions, can all be estimated in practice. In real world where  Assumption 10 is unverifiable, we estimate $T$ by plugging in the observed label priors and empirical label proportions. The real-world adapted algorithms presented in algorithms 2 and 3 in the revision work regardless of Assumption 10. The details of this plug-in approach are discussed in section 5.4 in the revision.

---

> > > ### Comment · Reviewer_tEEw · 2022-08-10
> > > **Thanks for the discussion**
> > >
> > > Thanks for the further discussion. My original question actually means that, in practice, we may have a very large number of bags & classes, and annotating each bag with label proportions could be difficult ($\gamma_k$ may not be observed); how to estimate the multi-class noise matrix in such cases? In the binary case, this question relates to how to estimate the mixing proportions in the mutually contaminated model which was previously studied in Classification with asymmetric label noise: Consistency and maximal denoising, COLT 2013.

---

### Official Review · Reviewer_vSZS · 2022-07-18

**Rating:** 7
**Confidence:** 2
**Soundness:** 3 good
**Presentation:** 3 good
**Contribution:** 3 good

**Summary:**

The paper investigates multiclass learning from label proportions, i.e. the situation in which the training data is only available in "bags". For each bag, the fraction of samples belonging to a certain class is known, but the exact assignment of instances to classes is not. The idea is to assign an arbitrary label to each instance in a bag, and then consider the bag to be a noisy learning problem with known transition matrix. For learning with noisy labels, they employ the forward-correction approach, proving new calibration-style generalization bounds for this, and then extending these to the multiple-noise matrix (i.e. multiple bags -- learning from label proportions case).

**Questions:**

In equation (1), shouldn't $\alpha_i(c)$ also be an empirical/estimated quantity?

l. 713 How de we know $A(\epsilon)$ is the closed interval, as opposed to being open?

Proof of Thm. 17
Where does the first line, $\mathcal{C}_2 < \delta(\mathcal{C}_1) \implies \ldots $, come from?

Lemma 22 should require that the matrix be invertible, I think.

**Limitations:**

Not a limitation per se, but as far as I can see this work leaves open quite trivial imporvements of the proposed algorithm, i.e. choosing the labels that are assigned to the bags in a way that minimizes the amount of noise that needs to be assumed.

**Strengths And Weaknesses:**

While the idea of interpreting learning from label proportions as a noisy-label-learning problem is not original to this paper, the multiclass application and use of forward-correction instead of unbiased loss functions constitute a non-trivial step over the existing literature. I agree with the authors that the additional results for forward correction may be of independent interest, though whether people working on noisy-label-learning will find these results in this paper I don't know. Maybe these results should, in addition to being stated here, be presented at some workshop or similar.

Overall, I found the paper to be well written and easy enough to follow in the main argument. In terms of notation, maybe this could be improved a bit, e.g. the symbol $\gamma_{k_{i, c_1}}$ has 3(!!) levels of subscripts. Similarly, the acronyms can get rather long:
LLPFC-..., LLPGAN, LLPVAT, LMNTM. None of these are actually pronouncable. Given that the context of the paper is LLP, I think it would be enough to mark the algorithms as FC-..., GAN, VAT, for example.

So far, I've only gone through the math in appendices C1 and C2. Aside from the comments below, this looks good. I will update the review once I've managed to go through the rest of the proofs.

---

> ### Author Response · Authors · 2022-08-02
> **Author Response**
>
> Thank you for the detailed review.
>
> # Questions:
>
> ## 1. In equation (1), shouldn't $\alpha_i(c)$ also be an empirical/estimated quantity?
>
> In equation (1), $\alpha_i(c)$ is assumed to be known for theoretical analysis. In practice, $\alpha_i(c)$ is replaced by its estimate $\hat{\alpha}_i(c)$ as a plug-in fashion. The estimation of $\alpha_i(c)$ is discussed on line 323-330 in the original submission and line 286-294 in the revision.
>
> ## 2. l. 713 How do we know $A(\epsilon)$ is the closed interval, as opposed to being open?
>
> Thank you for your questions.
>
> Here's a quick proof to show $A(\epsilon) = [0, \delta(\epsilon)]$.
>
> Proof: By lemma 15, $\exists \delta > 0$ such that $\delta \in A(\epsilon)$. So $\delta(\epsilon) > 0$.
>
> Note that if $ 0 < \delta_1 < \delta_2$ and $\delta_2 \in A(\epsilon)$, then $\delta_1 \in A(\epsilon)$. (*)
>
> Fix $\delta \in [0, \delta(\epsilon))$, then $\exists \delta' \in (\delta, \delta(\epsilon)]$ such that $\delta' \in A(\epsilon)$ because $\delta(\epsilon) = \sup A(\epsilon)$. By (*), this $\delta \in A(\epsilon)$.
>
> So $[0, \delta(\epsilon)) \subset A(\epsilon)$. (**)
>
> Fix $x\in\mathcal{X}$ and $q \in \Delta^C$ such that $C_{l_T, P_T, x} (q) - C_{l_T, P_T, x}^* < \delta(\epsilon)$. Then $\exists \delta'' \in \mathbb{R}$ such that $C_{l_T, P_T, x} (q) - C_{l_T, P_T, x}^* < \delta'' < \delta(\epsilon)$. By (**), $\delta'' \in A(\epsilon)$. So $C_{L, p, x} (q) < C_{L, P, x} ^ * + \epsilon$.
>
> Since the choices of $x$ and $q$ are arbitrary, we have $\delta(\epsilon) \in A(\epsilon)$.
>
> ## 3. Proof of Thm. 17 Where does the first line, $C_2 < \delta(C_1) \implies \dots$, come from?
>
> We can use $\delta(\epsilon) \in A(\epsilon)$ directly by replacing $\epsilon$ with $C_{1, x}(q)$. $\delta(C_{1, x}(q)) \in A(C_{1, x(q)})$ , so $C_{2, x}(p) < \delta(C_{1, x}(q)) \implies C_{1, x}(p) < C_{1, x}(q)$ by  the definition of $A$.
>
> ## 4. Lemma 22 should require that the matrix be invertible, I think.:
>
> It's been said that ``Suppose that M is non-singular'' on line 778-779 in the appendix in the original submission as well as line 750-751 in the revision.
>
> # Limitations:
>
> To optimize the weights or the assignment of bags we would need to optimize the composition of our two bounds:
> $\theta ( R_{L, P} (f)  - R_{L, P}^* ) \leq Empirical Risk + Generalization Error Bound - R^*_{l, P, T}$. This is in contrast to the approach with backward correction [Scott and Zhang 2020] which does not require the excess risk bound (because their excess target risk is simply proportional to the excess surrogate risk). Therefore, to optimize the composition of our bounds, we'd need to estimate the surrogate Bayes risk, a challenging task. We also note that both the generalization error bound and excess risk bound involves weights $w_i$ and noise matrices $T_i$. Therefore, even if the surrogate Bayes risk were somehow known, the resulting integer programming problem is much more involved than for the backward correction, where it's a simple matching problem. For example, with the cross-entropy loss and using our bound in Proposition 6 and 12 in the original submission (corresponding to proposition 11 and 22 in the revision, respectively), we would need to optimize for the following bound:
>
> $ R_{L, P}(f) -  R_{L, P}^* \leq \sqrt{\frac{1}{\sum_{i} \frac{w_i}{2||T_i^{-1}||^2}}} \sqrt{Empirical Risk - {R}^*_{l, P, {T}} + (A  + B - D \min_{i, c} \log (\frac{1}{C} \sum_{j=1}^C T_i(c, j)))\sqrt{\sum_{i,c}\frac{w_i^2}{n_{i,c}}}} $
> where $A$, $B$, and $D$ are positive constants depending on the function space, the number of classes $C$, and $\delta$. $T_i(c,j)$ is the entry at $c$-th row and $j$-th column in the matrix $T_i$.
>
> Fortunately, LLPFC with random partitioning and uniform weights demonstrates superior empirical results in the experiments and outperforms other multiclass LLP methods by a significant margin. We believe weight optimization is much more important for the backward correction, where the loss functions can have large and disparate magnitudes (which need to be offset by carefully chosen weights), than it is for forward correction where the outputs of the inverse link function are in the unit simplex and thus all of a comparable magnitude. A similar point is made by [Patrini et al. 2017](https://openaccess.thecvf.com/content_cvpr_2017/papers/Patrini_Making_Deep_Neural_CVPR_2017_paper.pdf) in the last two sentences in the first paragraph of section 6.

---

### Meta-Review · Area_Chair_dDVs · 2022-08-25

**Recommendation:** Accept
**Confidence:** Certain

**Metareview:**

This paper proposes an approach to LLP based on a reduction to learning with label noise, using the forward correction (FC). The method is verified both theoretically and experimentally. The theory of the FC loss may also be of independent interest. The authors have addressed most of the comments. After the discussion, the reviewers unanimously recommended acceptance.

**Award:**

No

---

### Decision · Program_Chairs · 2022-09-14

Accept